# Encapsulated Papillary Carcinoma: A Rare Case Report and Its Imaging Features

**DOI:** 10.3390/diagnostics12092098

**Published:** 2022-08-30

**Authors:** Noorzuliana Ahmad, Arasaratnam A. Shantini, Iqbal Hussain Rizuana, Muhammad Rohaizak

**Affiliations:** 1Department of Radiology, Faculty of Medicine, Universiti Kebangsaan Malaysia, Cheras, Kuala Lumpur 56000, Malaysia; 2Department of Radiology, Hospital Kuala Lumpur, Jalan Pahang, Kuala Lumpur 50586, Malaysia; 3Department of Surgery, Faculty of Medicine, Universiti Kebangsaan Malaysia, Cheras, Kuala Lumpur 56000, Malaysia

**Keywords:** papillary breast lesion, encapsulated papillary carcinoma, mammography

## Abstract

Papillary lesions in the breasts are uncommon and have a wide range of pathologies. Due to diverse non-specific findings radiologically and histologically, papillary neoplasms are always a challenge to radiologists. Encapsulated papillary carcinomas (EPCs) of the breast, also known as intracystic papillary carcinomas, are a subgroup of intraductal papillary lesions of the breast. We present a case of a painless right breast lump with the aim to describe a rare encapsulated papillary carcinoma and its imaging features.

A 66-year-old Indian lady with dyslipidemia was referred to the Surgical Clinic after complaining of a right painless breast lump that she had just noticed for a week and was progressively growing. There was no nipple discharge or skin changes. On examination, there was a 3 cm × 3 cm lump at the right breast upper outer quadrant with hard consistency and fixed to the skin. There was no abnormality detected on the left breast.

A mammogram study was requested, and it showed a lobulated nodule with irregular margin, mild spiculation and coarse calcification in the right mid-outer quadrant (Figure 1). Complimentary ultrasound showed a heterogenous mixed solid cystic lesion with irregular margins on the right 9 o’clock (Figure 2). This nodule did not demonstrate significant Doppler signals. A diagnosis of BIRADS 4 category led to an ultrasound-guided biopsy on the same day.

Histopathological evaluation (HPE) of the biopsy revealed a right papillary neoplasm. The patient was counseled for excision of the lesion, and she agreed to right wide local excision (WLE). She was also informed and understood about the possibility of a second surgery involving the right breast. The HPE of the surgery came out as right encapsulated papillary carcinoma (Figure 3) with estrogen (ER), progesterone (PR) and HER2 receptors positive. Unfortunately, the margins were involved with focal infiltration on HPE. Therefore, she then underwent a second surgery; right mastectomy with right axillary clearance (MAC) a month later.

Other than that, her bone scan and computed tomography (CT) for staging revealed no evidence of distant metastasis. The patient already completed oral hormonal therapy with tablet anastrozole and is currently well under routine follow-up by the surgical team.

Papillary lesions are uncommon and can be broadly characterized as benign or malignant [1,2,3]. They can be benign such as in solitary and multiple intraductal papillomas and atypical ductal hyperplasia (ADH) within a papilloma. They are classified as malignant ductal carcinoma in situ (DCIS) arising in papilloma, papillary DCIS, solid or invasive papillary carcinoma and intracystic or encapsulated papillary carcinoma (EPC) [1,3].

EPC is defined by the presence of papillary carcinoma in a cystically dilated duct [1,3]. It is a rare form of breast malignancy that accounts for about 0.5–1% of all breast malignancies [2,3,4,5]. It is more common in postmenopausal females and its incidence is reported less commonly among Asians. A patient with EPC usually presented with a painless breast lump with or without nipple discharge. The breast sometimes will be swollen due to a huge cystic mass [2,3,4]. There are cases where it is an incidental finding on a mammogram of an asymptomatic patient. Further evaluation with imaging and biopsy is always needed to confirm the diagnosis. Histologically, EPC is described as an expansile papillary lesion that is surrounded by a thick wall and an absent myoepithelial cell (MEC) lining. Their absence can be verified by immunohistochemical stains, thus differentiating the EPC from other types of papillary lesions [5].

Our patient’s mammography showed an irregular, mildly spiculated, lobulated mass in her right mid-outer quadrant associated with coarse calcification. Her complementary ultrasound revealed a heterogenous mixed, solid cystic mass on her right breast with irregular margins and echogenic debris within. No significant vascularity. The mass arises suspicion (BIRADS 4) and a biopsy was needed for confirmation.

Imaging-wise, papillary lesions also demonstrate various findings. In mammography, the presence of a growing lesion that is round, oval, or lobulated in shape, with usually well-defined margins, raises suspicion of papillary carcinoma [1,4,6,7]. The lesion may or may not have microcalcifications appearing within the tumor [6,8]. Spiculation, however, is not usually seen [3]. However, in our patient, there was mild spiculation seen on the mammogram. Many studies conclude that spiculation is uncommon [3,9,10,11]. Hernandez et al. reported although speculation is present, this is not always due to invasion but could be related to sclerosis and inflammation.

Ultrasonography is used to distinguish the cystic from the solid variety of papillary carcinoma. On ultrasound, the EPCs may have a hypoechoic, anechoic or mixed appearance with respect to their solid or cystic component. Possible internal vascularity can be seen with Doppler imaging [1,3,6]. The presence of vascular signals indicates a vascular lesion, and this lesion tends to bleed. Thus, the hemorrhagic lesion may appear with fluid-debris level within the cyst [6]. These findings have overlapping features with the benign papilloma, thus making it difficult to characterize the lesion with imaging alone [3,8].

The appearance of complex cystic lesions in ultrasound also can give rise to other differentials such as breast abscess, fat necrosis and hematoma. Both sonographic and mammographic findings are not sensitive nor specific to papillary lesions. Nonetheless, imaging is still important for the clinician to guide tissue biopsy as well as to stage the lesion.

It is debatable how to manage papillary lesions. Some clinicians agree that not all papillary lesions should be excised if a core biopsy shows a benign lesion. However, core biopsies have pitfalls. A core biopsy might miss the region with an absence of MEC lining, which histologically differentiates EPC from other lesions [8,9]. Fine needle aspirations also encounter a similar issue such as focal carcinoma in situ being missed [10,11]. In our center, the main therapeutic procedure so far is surgical excision and after careful consideration of the risks and benefits, hormonal therapy may be added. There is currently no role for cytotoxic chemotherapy [12]. In our patient, wide local excision was done initially. However, as the HPE came back as a malignant tumor with margins involved and focal invasion, a right MAC was done. Hormonal therapy was also added in view of the triple positivity of the ER, PR and HER2 receptors [13,14].

## Figures and Tables

**Figure 1 diagnostics-12-02098-f001:**
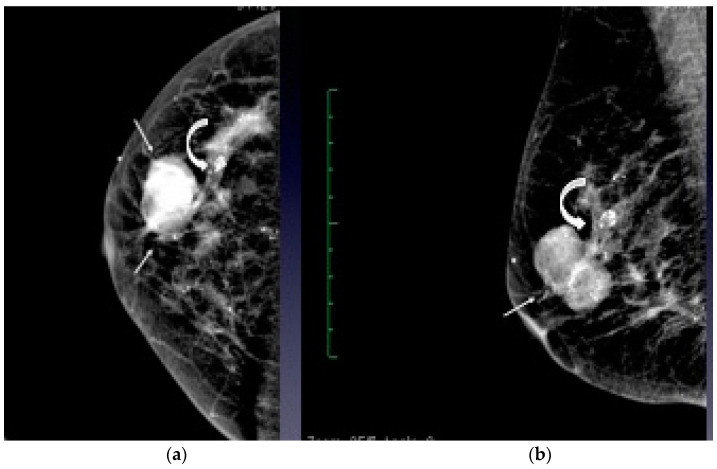
Craniocaudal (**a**) and mediolateral oblique (**b**) mammograms of the right breast show a lobulated mass with irregular margin, mild spiculation and coarse calcification (curved arrow) in the right mid-outer quadrant.

**Figure 2 diagnostics-12-02098-f002:**
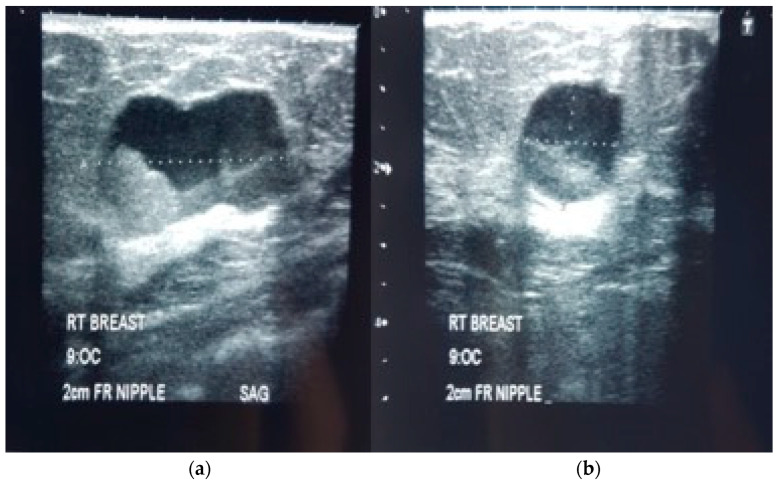
Sagittal (**a**) and axial (**b**) ultrasound views of the right breast show a heterogenous mixed solid cystic lesion with an irregular margin of the solid component at the 9 o’clock (OC) position.

**Figure 3 diagnostics-12-02098-f003:**
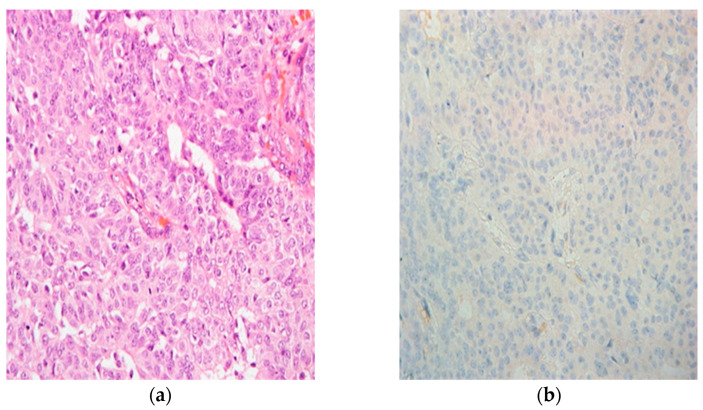
(**a**) Papillary structures with surrounding fibrous capsule (haematoxylin and eosin, H&E, ×400). (**b**) Tumor cells show negative uptake of stain (p63, ×400).

## Data Availability

Not applicable.

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
