# Peer review of "Encapsulated Papillary Carcinoma: A Rare Case Report and Its Imaging Features"

_diagnostics, 2022, doi:10.3390/diagnostics12092098_

Round 1
Reviewer 1 Report
This is a well-presented case report on an encapsulated papillary carcinoma presenting as a painless right breast lump in a 66-year-old lady. Images and references are OK, as well as the literature review. A minor, professional English language review is advised.
Author Response
Dear Reviewer 1,
We thank the reviewer very much for reviewing the manuscript and for your comments and suggestions. We have made every attempt to address the comments. We have taken every effort to improve on our English.
Best regards,
Authors

Reviewer 2 Report
Thank you for this interesting casa report. I'm not sure if right axillary treatment was the best think to do
Author Response
Reviewer's comment:
Thank you for this interesting case report. I'm not sure if right axillary treatment was the best thing to do
Author's reply:
Dear Reviewer,
We thank the reviewer very much for reviewing the manuscript and for their positive response.
We have revised the manuscript accordingly.
Rational for right axillary treatment:
There was presence of focal infiltration seen on HPE of the 1st surgery for which treatment of choice in our country is sentinel lymph node or axillary dissection. Unfortunately, sentinel lymph node was not available at the centre where the patient initially seeked treatment, the surgeon there opted for level 2 axillary dissection.
Best regards,
Authors